



# ClimKern v1.1.2: a new Python package and kernel repository for calculating radiative feedbacks

Tyler P. Janoski[1-5], Ivan Mitevski[6], Ryan J. Kramer[7], Michael Previdi[2], and Lorenzo M. Polvani[1-2,8]

[1]Dept. of Earth and Environmental Sciences, Columbia University, New York, NY, USA
[2]Lamont-Doherty Earth Observatory, Columbia University, Palisades, NY, USA
[3]NOAA Center for Earth System Science and Remote Sensing Technologies (CESSRST-II), New York, NY, USA
[4]City College of New York, New York, NY, USA
[5]NOAA National Severe Storms Laboratory, Norman, OK, USA
[6]Dept. of Geosciences, Princeton University, Princeton, NJ, USA
[7]NOAA Geophysical Fluid Dynamics Laboratory, Princeton, NJ, USA
[8]Dept. of Applied Physics and Mathematics, Columbia University, New York, NY, USA

**Correspondence:** Tyler P. Janoski (tjanoski@ccny.cuny.edu)

**Abstract.**

Climate feedbacks are a significant source of uncertainty in future climate projections and need to be quantified accurately and robustly. The radiative kernel method is commonly used to efficiently compute individual climate feedbacks from climate model or reanalysis output. Despite its popularity, it suffers from complications, including difficult-to-locate radiative kernels,

inconsistent kernel properties, and a lack of standardized assumptions in radiative feedback calculations, limiting the robustness and reproducibility of climate feedback computations. We designed the ClimKern project to address these issues with a kernel repository and a separate but complementary Python package of the same name. We selected eleven sets of radiative kernels and gave them a common nomenclature and data structure. The ClimKern Python package provides easy access to the kernel repository and functions to compute feedbacks, sometimes with a single line of code. The functions contain helpful optional

parameters while maintaining standard practices between calculations.

After documenting the kernels and ClimKern package, we test it with sample climate model output to explore the sensitivity of feedback calculations to kernel choice. Interkernel spread shows considerable spatial heterogeneity, with the greatest spread in the Arctic and over the Southern Ocean. Considerable sensitivity to kernel choice is found even in the global means, with the surface albedo and cloud feedbacks showing the greatest spread across different kernels. Our results highlight the importance

of using more than one radiative kernel and standardizing feedback calculations, like those offered by ClimKern, in climate feedback, climate sensitivity, and polar amplification studies. As ClimKern continues to evolve, we hope others will contribute to its development to make it even more useful to the feedback community.



## 1 Introduction

One of the fundamental questions in climate science is how much the surface will warm in response to the radiative forcing
imposed by increasing $CO_2$ concentrations. A typical framework for answering this question is expressing the top-of-the-atmosphere (TOA) radiative imbalance, $\Delta R$, as

$$\Delta R = \Delta F + \lambda \Delta T, \tag{1}$$

where $\Delta F$ is the radiative forcing, $\lambda$ is the net climate feedback parameter, and $\Delta T$ is the global mean surface temperature response. The feedback parameter $\lambda$ is the increase in outgoing radiation per degree warming with units of W m$^{-2}$ K$^{-1}$ and
represents the effects of all global average radiative feedbacks combined. Using this forcing-feedback framework, we can compute the equilibrium climate sensitivity (ECS), which is the global mean surface temperature response needed to restore the TOA imbalance to zero after doubling $CO_2$, (Sherwood et al., 2020), as

$$\text{ECS} = \frac{\Delta F_{2 \times CO_2}}{-\lambda}. \tag{2}$$

The complexity of the climate system and observational uncertainty lead to large uncertainties in estimates of ECS, with the
climate feedback parameter $\lambda$ considered a greater source of uncertainty in ECS than the forcing $\Delta F$ (Sherwood et al., 2020). The uncertainty in $\lambda$ stems from the significant uncertainty in its components, notably the cloud and water vapor feedbacks (Roe and Baker, 2007; Andrews et al., 2012; Vial et al., 2013; Sherwood et al., 2020). Feedback uncertainty is also important on regional scales. For instance, the Arctic, which is warming faster than the global average in a phenomenon known as Arctic amplification (AA), is characterized by considerable feedback uncertainty, making it difficult to attribute warming to individual
feedbacks (Pithan and Mauritsen, 2014; Hahn et al., 2021; Shi and Lohmann, 2024).

The net feedback parameter can be linearly decomposed into a sum of individual feedbacks: $\lambda = \sum_i \lambda_i$, where $\lambda_i$ represents the contributions of individual feedbacks: lapse rate, Planck, water vapor, surface albedo, and cloud feedbacks. There are two caveats to this decomposition worth noting. First, representing $\lambda$ as a linear combination of individual feedbacks ignores the interaction *between* feedbacks, which can be important, especially on local scales (Feldl and Roe, 2013; Knutti and Rugenstein,
2015; Feldl et al., 2017; Huang et al., 2021). Second, $\lambda$ and its individual components are likely not constant, varying with the climate state and with the pattern of surface temperature change (Knutti and Rugenstein, 2015; Gregory and Andrews, 2016; Dong et al., 2019; Meyssignac et al., 2023). Even with these caveats, the linear decomposition of feedbacks remains a commonly used framework.

The most common way to calculate individual radiative feedbacks is by using radiative kernels (Soden et al., 2008). Radiative
kernels are the pre-calculated radiative sensitivities at some vertical level, often the TOA, to incremental changes in climate variables, such as temperature, water vapor, and surface albedo. The TOA radiative imbalance due to feedbacks, $\Delta R_\lambda$ (equivalent to $\lambda \Delta T$, see Eq. 1), is decomposed as

$$\Delta R_\lambda = \sum_i \frac{\partial R_i}{\partial x_i} \Delta x_i, \tag{3}$$



where $\frac{\partial R}{\partial x}$ is the radiative kernel, and $\Delta x$ is the change in a climate variable (e.g., following $2\times CO_2$). The radiative kernel
method offers several advantages over other methods of calculating radiative feedbacks. For example, radiative kernels can
be applied to virtually any gridded data (e.g., climate model output, reanalysis products, etc.) as long as standard variables,
such as temperature and specific humidity, are available. Using existing radiative kernels also alleviates the need to perform
computationally expensive partial radiative perturbation calculations or run offline radiative transfer models (Wetherald and
Manabe, 1988; Colman and McAvaney, 2011; Smith et al., 2020). Another use of radiative kernels is the decomposition of the
effective radiative forcing into individual components, allowing for the separation and quantification of specific adjustments,
such as changes in cloud properties or aerosol concentrations (Larson and Portmann, 2016).

The underlying assumption of the radiative kernel method is that differences between kernels produced using different models
are minor compared to differences in the climate responses between models. This is because interkernel variation stems only
from differences in radiative transfer models and model base states (Pincus et al., 2020), both of which are, ideally, physically
reasonable representations of the real world (Soden et al., 2008). This assumption of minor differences between kernels enables
intermodel feedback comparisons and allows for the use of virtually any radiative kernel to calculate feedbacks.

A question naturally follows: are the differences between kernels actually minor? Only a few studies have addressed this
question. Soden et al. (2008) found that among three kernels calculated using different models, zonal mean kernels varied
by $\sim 10\%$ except for in the Southern Ocean, where they varied by $\sim 30\%$. In the context of the global mean, temperature
and water vapor kernels only varied by $\sim 5\%$, although the surface albedo kernel varied considerably more. In a more recent
study, Hahn et al. (2021) found considerable spread in global and regional surface albedo and cloud feedbacks calculated from
different kernels. Huang and Huang (2023) documented a new set of kernels and found agreement in the global mean TOA
feedbacks among the seven kernels they considered but notable differences in the surface feedbacks. Although we do not seek
to completely answer the question of the importance of interkernel differences, here we note that given the popularity of the
radiative kernel method, it deserves more attention; however, the current research environment makes intercomparing different
radiative kernels difficult.

Using different kernels introduces uncertainty that can limit the reproducibility and robustness of climate feedback studies.
First, although many kernels have been produced since the early studies of Soden et al. (2008) and Shell et al. (2008), they are
scattered among different research groups and institutions, making them difficult to locate; even after accessing a kernel, there
is often little to no guidance on their proper usage. Second, kernels vary considerably in their properties, such as horizontal and
vertical grids, model tops, sign conventions, and nomenclature, which may introduce calculation discrepancies across studies.
Lastly, using kernels to calculate radiative feedbacks requires several choices and assumptions; examples include what base
temperature to use when calculating the specific humidity increase from a 1K increase in atmospheric temperature and how to
handle vertical integration to the surface while accounting for surface pressure and terrain. These three factors make comparing
results between feedback studies difficult, even when studies may use the same radiative kernels.



To standardize radiative feedback calculations and establish a central kernel repository, we created the ClimKern project. This project consists of two distinct parts: the ClimKern Python package, an open-source library for computing radiative feedbacks, and the ClimKern repository, which provides easy access to 11 sets of radiative kernels, as of this writing, computed from various climate models, reanalyses, and satellite observations. The package provides functions for calculating radiative feedbacks using any of the radiative kernels in just one or two lines of code per feedback. The package greatly enhances the reproducibility of feedback studies by standardizing the assumptions and choices. It also enables straightforward interkernel comparisons to better understand the role of kernel choice in these studies.

The remaining sections are organized as follows: section 2 provides detailed information about ClimKern radiative kernels and the sample data we included for demonstration purposes. Section 3 covers the methodological choices made in crafting the feedback calculation functions. Section 4 shows the results of using the package with the sample climate model output to calculate feedbacks. In section 5, we put our package and the sample results in the context of the greater climate feedback and sensitivity community.

## 2 Data

### 2.1 Radiative kernels

We acquired 11 sets of TOA radiative kernels that were either publicly available or made available to us by the creators. To be included in ClimKern, a kernel product must have 4-dimensional water vapor and air temperature kernels, as well as 3-dimensional surface temperature and surface albedo kernels. The kernels must be monthly averages to capture the seasonal variations in TOA radiative fluxes and must be on horizontal latitude-longitude grids. The above requirements were chosen to ensure ease of use and that feedback calculations using different kernels are directly comparable. In this first version of ClimKern, we excluded radiative kernels that require nontraditional (i.e., considerably different from Soden et al. (2008)) variables to compute feedbacks; examples include the cloud kernels from Zelinka et al. (2012), which require satellite-simulator produced output, and new kernels from the NASA Goddard Institute for Space Studies that use column precipitable water and sea ice fraction variables (Zhang, 2023). We also excluded band-by-band or "spectral" kernels, such as those in Bani Shahabadi and Huang (2014) and Huang et al. (2024).

Seven of the 11 TOA kernel sets had corresponding surface kernels for calculating radiative feedbacks from a surface perspective, as in Pithan and Mauritsen (2014) and Laîné et al. (2016). Although they are included in the repository for ease of access, surface feedback calculations have not been implemented in ClimKern, and our discussion exclusively focuses on TOA kernels and feedbacks. Future versions of ClimKern may expand compatibility to surface kernels and other kernel types.

Details about each kernel set can be found in Table 1. These 11 kernel sets were developed independently using a variety of data sources for their base states, including climate model output, reanalysis data, and satellite observations (Soden et al., 2008; Huang et al., 2017; Kramer et al., 2019). Horizontal resolutions range from several degrees to under one degree in latitude and




**Table 1.** The 11 radiative kernel sets included in the ClimKern repository. From left to right, the table contains the kernel names, the horizontal resolution, the number of vertical levels in the ClimKern version of the kernel, the highest pressure level of the kernel, and the paper that first documents the kernel.

| kernel name | horizontal resolution (lat×lon) | number of vertical levels | highest pressure level (hPa) | citing paper |
|---|---|---|---|---|
| BMRC | 3.2°×5.6° | 19 | 1 | Soden et al. (2008) |
| CAM3 | 2.8°×2.8° | 17 | 10 | Shell et al. (2008) |
| CAM5 | 0.94°×1.25° | 22 | 3.64 | Pendergrass et al. (2018) |
| CERES | 0.5°×1° | 30 | 0.1 | Thorsen et al. (2018) |
| CloudSat | 2°×2.5° | 17 | 10 | Kramer et al. (2019) |
| ECHAM6 | 1.88°×1.88° | 19 | 1 | Block and Mauritsen (2013) |
| ECMWF-RRTM | 2.5°×2.5° | 24 | 1 | Huang et al. (2017) |
| ERA5 | 2.5°×2.5° | 37 | 1 | Huang and Huang (2023) |
| GFDL | 2°×2.5° | 17 | 10 | Soden and Held (2006) |
| HadGEM2 | 1.25°×1.88° | 19 | 1 | Smith et al. (2018) |
| HadGEM3-GA7.1 | 1.25°×1.9° | 39 | 3 | Smith et al. (2020) |

longitude. Nearly all the kernels were already available on standard pressure levels, the desired vertical coordinate to ensure kernel compatibility with various climate model output to calculate feedbacks. Kernels available on their native model grids (i.e., CAM5 & HadGEM3-GA7.1) were linearly interpolated to pressure levels. The native CAM5 kernels were available on hybrid sigma-pressure vertical coordinates; isobaric levels in the upper troposphere were unchanged, while hybrid levels in the lower and mid-troposphere were converted to the standard pressure levels used in the Coupled Model Intercomparison Project Phase 6 (CMIP6) (Eyring et al., 2016). The native HadGEM3-GA7.1 kernels were on a pure sigma vertical coordinate that lacks isobaric surfaces. Because they were specifically developed with a high model top and enhanced vertical resolution to capture stratospheric adjustments (Smith et al., 2020), they were interpolated to 39 pressure levels, the highest standard CMIP6 vertical resolution. For further details about individual kernels, see the corresponding citing papers in Table 1.

After we collected the kernels and performed the necessary regridding, we combined each kernel set into one netCDF file per kernel source. The native kernel variables were renamed so as to have a standardized set of variables, and their units or other metadata were altered for consistency and accuracy. For example, all surface albedo kernels had their units changed to $Wm^{-2}\%^{-1}$ if needed. We then inspected the kernels to find inconsistencies with their sign conventions, which were corrected. The resulting dataset was uploaded to Zenodo (Janoski et al., 2024a), from where it can be downloaded manually or via a built-in script in the ClimKern Python package.



## 2.2 Sample climate model output

We also provide a tutorial dataset within the package to calculate, for verification purposes, the feedbacks given in Table 2. The package includes a function that accesses the sample data derived from pre-industrial and abrupt-2×$CO_2$ fully coupled runs
using the Large Ensemble version of the Community Earth System Model 1 (CESM1-LE). The CESM1-LE model incorporates the Community Atmosphere Model version 5 (CAM5) with 30 vertical levels and the Parallel Ocean Program version 2 (POP2) with 60 vertical levels. The model operates at a horizontal resolution of 1° across all components (Kay et al., 2015). These experiments have been extensively documented in prior studies (Mitevski et al., 2021, 2022, 2023). We also provide the effective radiative forcing (ERF) from these experiments, calculated from simulations with prescribed pre-industrial sea surface
temperatures and sea ice concentrations as in Forster et al. (2016). Lastly, we provide the instantaneous radiative forcing (IRF) for the same dataset calculated offline with the radiative transfer model SOCRATES (Edwards and Slingo, 1996; Manners, 2015). Note that because SOCRATES is not the radiative transfer scheme used in CESM, it may not yield perfect energy budget closure (e.g., following Eq. 1) even with correct kernel decomposition.

## 3 Feedback calculations

The ClimKern Python package, hereafter referred to as simply "ClimKern," contains many built-in functions for calculating radiative feedbacks and other valuable quantities of interest. Note that all output from feedback functions is in the form of the TOA radiative perturbations from the feedback in units of $Wm^{-2}$; if the user wishes to express feedback values as per unit temperature ($Wm^{-2}K^{-1}$), that is usually achieved by dividing by the surface temperature response. We avoided incorporating this step into the functions as there are other ways of expressing feedbacks, such as in the form of warming contributions
(Pithan and Mauritsen, 2014; Goosse et al., 2018; Previdi et al., 2020; Janoski et al., 2023), and the radiative perturbations in $Wm^{-2}$ are useful to calculating rapid adjustments to radiative forcing (Smith et al., 2018). Below, we document the required user input for the feedback calculation functions and provide details on their methodologies. This is not an exhaustive list of functions available in ClimKern, and specifics are subject to change in future versions. Still, we hope it will prove helpful to discuss the philosophy behind the design of each function.

### 3.1 Temperature feedbacks

Temperature feedbacks refer to the radiative perturbations at the TOA from changes in the surface and atmospheric temperatures. Traditionally, the total temperature feedback is decomposed into the Planck feedback (or Planck response, depending on the specific definition of "feedback" used) and the lapse rate feedback (Soden and Held, 2006; Bony et al., 2006; Soden et al., 2008). The Planck feedback is the radiative response to a vertically uniform temperature change of equal magnitude
to that of the surface; it is the most fundamental response of the radiative budget to a change in temperature, following the Stefan-Boltzmann law (Previdi et al., 2021). The lapse rate feedback differs in that it reflects the *deviation* from vertically uniform warming to quantify the radiative effects of an altered tropospheric lapse rate.



ClimKern provides the `calc_T_feedbacks` function that computes the tropospheric Planck and lapse rate feedbacks using user-provided 4-D air temperature and 3-D surface temperature and pressure fields from two climate model simulations: a control and a perturbed simulation, the difference of which is used to calculate the temperature response. In the tutorial data provided with ClimKern, these are a $1 \times CO_2$ and a $2 \times CO_2$ (relative to preindustrial levels) simulation, respectively. Reanalysis data can be used in a similar fashion by separating data into two time periods for comparison.

First, ClimKern checks the input to ensure its format is compatible, including checking the time dimensions and units; then, the function will either proceed, issuing a warning to the user if any assumptions are made for missing metadata, or return an error for major incompatibilities (e.g., not providing input in the form of an Xarray DataArray (Hoyer and Hamman, 2017)). If the user did not provide an optional model- or user-defined tropopause, ClimKern will create a tropopause defined as 100 hPa at the Equator and linearly increasing with the cosine of latitude to 300 hPa at the poles. It will also read in the user-selected temperature and surface temperature kernels from locally stored package data. Using the xESMF module (Zhuang et al., 2023), the kernels are horizontally regridded using bilinear interpolation with periodic boundary conditions to match the resolution of the input model data. We elected to horizontally regrid to the input data's resolution so that the user always receives output on the same horizontal grid as the input.

Following this setup, ClimKern creates a monthly climatology from the control simulation surface and atmospheric temperatures and subtracts it from the perturbed simulation fields, yielding a surface and air temperature response. The air temperature response is linearly interpolated to match the vertical kernel resolution; subsequent testing for tropospheric feedbacks at the TOA demonstrates little difference if the input vertical resolution is used (not shown). Layer thicknesses are then calculated to be used in the vertical integration of the temperature feedbacks in the subsequent step. Note that the user-supplied perturbed simulation pressure and tropopause height are used when calculating the layer thicknesses to ensure that the vertical integration only extends from the surface to the tropopause.

The total air temperature response is decomposed into a vertically uniform component and a deviation therefrom to calculate the Planck and lapse rate feedbacks separately. Both feedbacks are calculated by multiplying the respective temperature response component, temperature kernel, and layer thickness array and taking a sum along the pressure axis. In the case of the Planck feedback, the surface temperature response is multiplied by the surface temperature kernel and added to this sum. The function then returns both feedbacks. To the user, all of this culminates in two lines of code:

```
import climkern as ck
LR,Planck = ck.calc_T_feedbacks(ctrl.T,ctrl.TS,ctrl.PS,
                                pert.T,pert.TS,pert.PS,pert.TROP_P,kern="GFDL")
```

where LR and Planck are the vertically integrated, monthly- and spatially varying lapse rate and Planck feedbacks, respectively, ctrl and pert are Xarray Datasets (Hoyer and Hamman, 2017) holding the control and perturbed simulations, T is the 4-dimensional air temperature, TS is the 3-dimensional surface temperature, PS is the 3-dimensional surface pressure, TROP_P



is the 3-dimensional tropopause height, and kern is the optional kernel choice argument. All feedback calculations share this kern argument, which defaults to "GFDL", to specify which of the 11 kernels ClimKern should use.

This function contains several options, including the kernel name, tropopause heights, all-sky or clear-sky feedbacks, and whether to calculate the feedbacks using relative humidity as a state variable, as in Held and Shell (2012). Further details about the computations and optional parameters can be found in the source code located in Janoski et al. (2024b).

**3.2   Water vapor feedback**

ClimKern also offers a `calc_q_feedbacks` function to compute water vapor feedbacks:

```
q_lw,q_sw = ck.calc_q_feedbacks(ctrl.Q,ctrl.T,ctrl.PS,
                                pert.Q,pert.PS,pert.TROP_P,
                                kern="GFDL",method=1)
```

where q_lw and q_sw are the TOA radiative perturbations from the longwave and shortwave water vapor feedbacks, respectively, Q is the 4-dimensional specific humidity, and all other variables are as they are in Section 3.1. Note that a "control" air temperature variable is required because water vapor kernels are traditionally calculated not using a unit increase in specific humidity but rather the specific humidity change corresponding to a 1K increase in temperature with constant relative humidity (Shell et al., 2008); consequently, the units of the water vapor kernels are $Wm^{-2}K^{-1}$.

The basic flow of the function is similar to that of the temperature feedbacks: first, ClimKern checks all input data and tries to identify proper units. If the user did not provide a DataArray with tropopause pressure, ClimKern constructs a default one. Next, ClimKern produces a monthly climatology of the control simulation surface pressure, specific humidity, and air temperature, masking values below the surface. ClimKern then computes the specific humidity response using a linear or logarithmic approach according to the method argument discussed below. As for the temperature feedbacks, the kernels are
regridded to the horizontal grid of the input data while the climatologies of the specific humidity and air temperature and the specific humidity response are put on kernel pressure levels. The product of the kernel, specific humidity response, and layer thickness is vertically integrated over the troposphere; a normalization factor must also be included, as discussed below.

Using a temperature perturbation to produce the water vapor kernels requires the water vapor kernels to be normalized by the change in specific humidity per unit temperature before calculating the feedback itself. Ideally, one would use the change
in specific humidity per unit temperature from the kernel calculation to normalize the kernel, as in Shell et al. (2008) and Pendergrass (2019), but this quantity is rarely included with the distributed kernels. Given that we have little information about the base states used in the individual kernel calculations, ClimKern utilizes the climatological air temperature from the user-provided control simulation to produce a water vapor kernel normalization factor using the Buck (1981) empirical formula for saturation vapor pressure.

The `calc_q_feedbacks` function also contains a unique "method" parameter that accepts one of four numeric arguments (1-4). Past studies vary in the way they compute the water vapor feedback - namely, whether to use the natural logarithm





of water vapor concentration and, if so, whether the log is approximated as the fractional change in water vapor. It is most common to use the natural log of specific humidity in water vapor feedback calculations because the absorption of longwave radiation by water vapor is roughly proportional to the logarithm of its concentration (Shell et al., 2008; Lacis et al., 2013; Colman and Soden, 2021); however, we also included an option to use the linear change in specific humidity, as in Pendergrass (2019). The difference in natural logarithms of specific humidity has sometimes been approximated as the fractional change in specific humidity. ClimKern includes that option for the specific humidity response and the normalization factor. The options and numeric arguments are:

1. Uses the actual logarithm for both the specific humidity response and the normalization factor.

2. Uses the fractional change approximation of logarithms only in the normalization factor, with the actual logarithm used in the specific humidity response.

3. Uses the fractional change approximation of logarithms in the specific humidity response & normalization factor.

4. Uses the linear change in specific humidity for both.

The function defaults to option 1. Further details can be found in the function's docstring (Janoski et al., 2024b).

## 3.3 Surface albedo feedback

The `calc_alb_feedback` function, which computes surface albedo feedback, is relatively straightforward; it requires the user to provide the upwelling and downwelling shortwave radiation at the surface from the control and perturbed simulations. The first step is to compute the surface albedo as the ratio of surface upwelling to downwelling radiation while masking areas with a downwelling radiation value of $0\,\mathrm{Wm^{-2}}$. ClimKern then takes the difference between the perturbed simulation's albedo and the control simulation's monthly climatological albedo. The desired albedo kernel is loaded from memory, regridded to the input horizontal resolution, and multiplied by the albedo response to produce the surface albedo feedback. As in the other ClimKern feedback functions, users may specify the kernel to use and whether to compute the all-sky or clear-sky feedback.

## 3.4 Cloud feedbacks

Cloud feedbacks are comparatively more complicated than the other feedbacks, owing to nonlinearities in kernel computations and the vertical overlapping of clouds (Soden and Held, 2006; Soden et al., 2008; Shell et al., 2008). Consequently, most traditional kernel sets do not include explicit shortwave or longwave cloud kernels, requiring alternate methods for calculating cloud feedbacks — most commonly, the residual and adjustment methods. ClimKern contains a function for each method, which we will detail below. Note that for both methods, ClimKern requires one or more radiative forcing terms that will vary with the given experimental setup (i.e., control and perturbation simulations). In other words, there is not a precise type of radiative forcing quantity, including ERF and IRF, that will suit every scenario. ClimKern avoids making assumptions regarding the forcing, and it is up to the user to ensure that all terms in the radiative budget are being properly accounted for. In our sample results (Section 4), we use the ERF to compute cloud feedbacks.





### 3.4.1 Residual method

In the residual method, the cloud feedbacks are computed as a residual of the TOA energy budget, that is:

$$\Delta R_{cloud} = \Delta R_{all-sky} - \Delta F - \sum_i \Delta R_i \qquad (4)$$

where $\Delta R_{cloud}$ is the TOA radiative perturbation from the cloud feedback, $\Delta R_{all-sky}$ is the net TOA radiative imbalance, $\Delta F$ is the radiative forcing (e.g., from $CO_2$), and $\sum_i \Delta R_i$ is the sum of the TOA radiative perturbations from other non-cloud feedbacks (Soden and Held, 2006; Zhang et al., 2018; Zhu et al., 2019). Put another way, the cloud feedback is assumed to be the missing piece in the TOA radiative budget after accounting for other terms. Although this method provides a "clean" approach that fully closes the radiative budget in a kernel feedback decomposition, it carries two main drawbacks. First, it is highly sensitive to uncertainties in the other terms and, especially, in the often unavailable radiative forcing $\Delta F$ (Soden et al., 2008). Second, since the cloud feedbacks are assumed to close the radiative budget, feedback decompositions using this method yield no separate error estimate, which is sometimes useful in evaluating radiative kernels. Despite these disadvantages, the residual method is still widely used.

ClimKern contains separate `calc_cloud_LW_res` and `calc_cloud_SW_res` functions to calculate the longwave and shortwave cloud feedbacks, respectively. For the longwave, ClimKern requires net longwave radiative flux at the TOA from the control and perturbed simulations, the longwave all-sky radiative forcing, and the radiative perturbations from the total temperature and longwave water vapor feedbacks. The shortwave function instead requires the net shortwave radiative flux at the TOA in the control and perturbed simulations, the shortwave all-sky radiative forcing, and the radiative perturbations from the surface albedo and shortwave water vapor feedbacks. From there, the cloud feedback is computed using Eq. 4 in both functions.

### 3.4.2 Adjustment method

The adjustment method for calculating cloud feedbacks is named as such because the change in cloud radiative effect (CRE) is "adjusted" for masking by other feedbacks and the radiative forcing to produce a cloud feedback:

$$\Delta R_{cloud} = \Delta CRE + \sum_i \Delta R_{i,clear-sky} - \Delta R_{i,all-sky} + (\Delta F_{clear-sky} - \Delta F_{all-sky}). \qquad (5)$$

where $\Delta CRE$ is the CRE response, $\Delta R_{i,clear-sky}$ and $\Delta R_{i,all-sky}$ are the clear-sky and all-sky radiative feedbacks, and $\Delta F_{clear-sky}$ and $\Delta F_{all-sky}$ are the clear-sky and all-sky radiative forcings (Soden et al., 2008; Zhang et al., 2018). $\Delta CRE$ is computed as $\Delta R_{all-sky} - \Delta R_{clear-sky}$, i.e. the difference in the all-sky and clear-sky TOA radiative flux. The adjustment method is considered less sensitive to uncertainties in the other terms, especially the forcing term (Soden et al., 2008). Additionally, since the resulting cloud feedback is not computed as a residual, it allows one to separately quantify the error in closing the TOA radiative budget.

The longwave and shortwave adjustment-method cloud feedbacks can be computed via the `calc_cloud_LW` and `calc_cloud_SW` functions. The longwave function uses as input the change in the longwave CRE, along with the all-sky and clear-sky radiative




perturbations at the TOA from the total temperature feedback, longwave water vapor feedback, and longwave radiative forc-
ing. The shortwave function uses the shortwave versions of the longwave function input, except that it uses the surface albedo
feedback instead of the temperature feedback. ClimKern includes separate `calc_dCRE_LW` and `calc_dCRE_SW` functions
that evaluate the change in longwave and shortwave CRE and that require several radiative fields from the user, including the
TOA all-sky and clear-sky LW or SW radiative fluxes in the control and perturbation simulations. After reading in all the
necessary input, the adjustment method cloud feedback functions calculate the differences between the all-sky and clear-sky
perturbations from non-cloud terms and combine them with the change in CRE to return the desired cloud feedback.

## 3.5 Other functions

We included several other utility functions in ClimKern. First, there are stratosphere versions of the temperature and water
vapor feedback functions, named `calc_strato_T` and `calc_strato_q`, respectively. They are mostly analogous to their
tropospheric counterparts, but the vertical integration is performed from the tropopause to the TOA. Next, ClimKern provides
a `calc_RH_feedback` function to calculate the relative humidity feedback following Shell et al. (2008), Held and Shell
(2012), and Zelinka et al. (2020). Typically, the relative humidity feedback would be a component of a radiative feedback
decomposition if the user calculated the temperature feedbacks with the `fixRH` option. Finally, the `spat_avg` function
computes the spatial average of a DataArray while weighting for the cosine of latitude. We refer the reader to Janoski et al.
(2024b) for additional documentation.

## 4 Results with sample data

### 4.1 Radiative kernels

Having outlined the data and functions packaged with ClimKern, we now focus on the characteristics of the TOA radiative
kernels. Fig. 1 shows the annual- and zonal-average mean kernel values and two across-kernel standard deviation ranges after
linearly interpolating to a common 17 standard pressure levels. Each kernel exhibits remarkably different spatial patterns in the
kernel mean and standard deviations and even, in some cases, between the all-sky and clear-sky versions of the same kernel.
The mean and standard deviation of the all-sky air temperature kernels (Fig. 1a) have two local maxima, one in the equatorial
upper troposphere and the other in the mid-to-high-latitudes lower troposphere in the Southern Hemisphere (SH). In the clear-
sky kernels, the lower tropospheric maximum is located over the Equator rather than the extratropical SH (Fig. 1b). Because the
all-sky and clear-sky kernels differ only by the existence of cloud effects in their calculations, the different maxima locations are
likely a result of clouds, which exert considerable influence on temperature kernels via cloud-top height temperatures (Kramer
et al., 2019). Overall, the clear-sky air temperature kernels have considerably less spread than the all-sky kernels (Fig. 1a-b),
highlighting the uncertainty introduced by clouds in radiative schemes.

The longwave water vapor kernels (Fig. 1c-d) do not appear to show as large sensitivity to clouds as the air temperature
kernels, with the exception of the deep tropics between 800 and 400 hPa. The longwave water vapor kernel mean is largest in



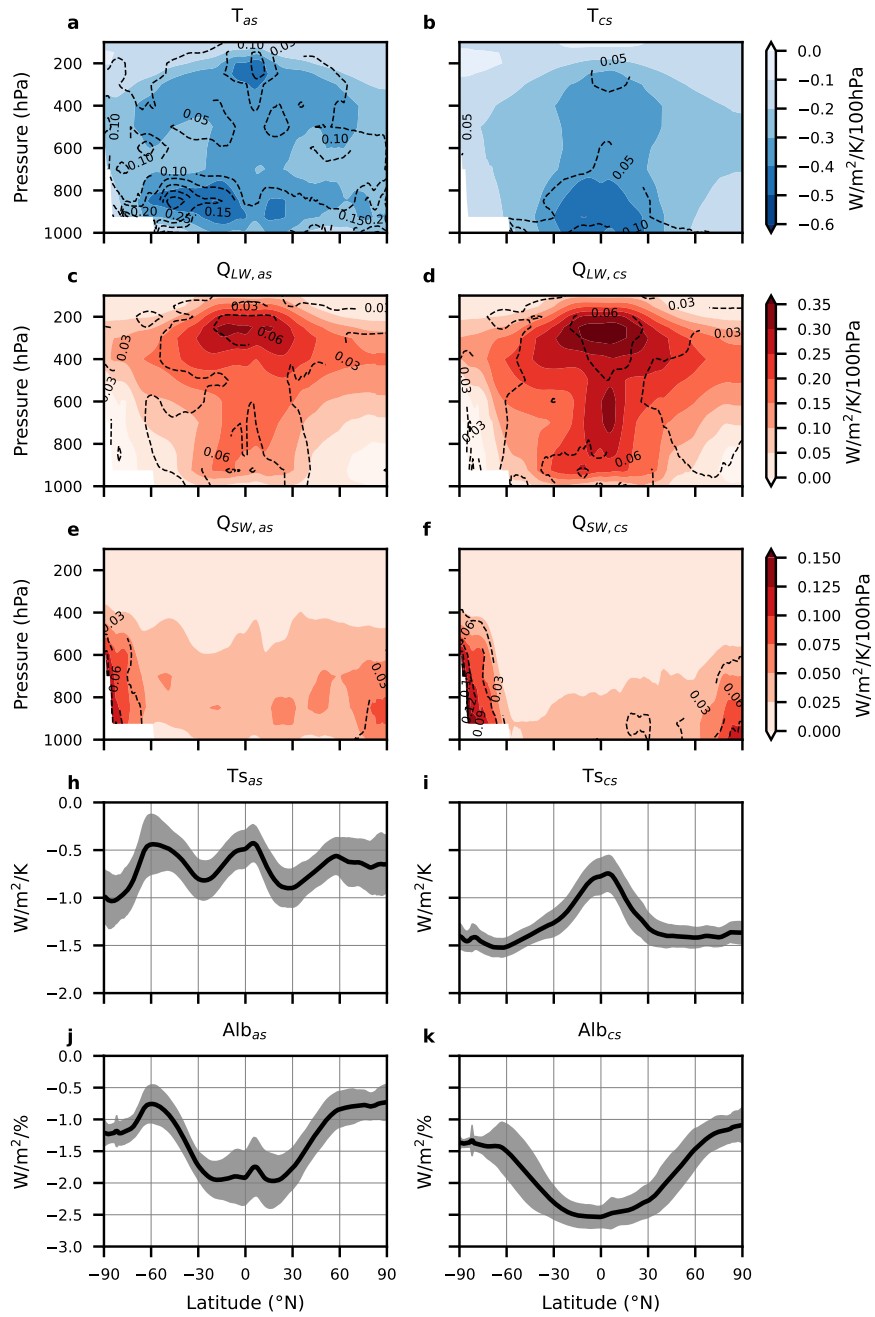

**Figure 1.** (a-b) The mean (shaded) and twice the standard deviation (contoured, dashed) of the all-sky (left) and clear-sky (right) temperature kernels, representing the response to a 1K increase in temperature. (c-d) as in (a-b), but for the longwave water vapor kernels, which reflect specific humidity changes associated with a 1K warming and fixed relative humidity. (e-f) as in (a-b), but for the shortwave water vapor kernels. (h-i) The mean (solid line) and two standard deviations (shading) of the all-sky (left) and clear-sky (right) surface temperature kernels. (j-k) as in (h-i), but for the surface albedo kernels, corresponding to a 1% increase in surface albedo.





the Equatorial upper troposphere and decreases with latitude, consistent with the findings of Huang et al. (2007). The pattern of the standard deviation mostly follows that of the mean with Equatorial maxima in both the high and low troposphere (Fig. 1c-d), indicating that this region is particularly sensitive to the base state and physics used in kernel production. The shortwave water vapor kernels (Fig. 1e-f) exhibit an increase in mean and standard deviation with latitude, opposite to that of the longwave kernels. As suggested by Huang and Huang (2023), the higher shortwave reflectivity of land and ice surfaces vs. ocean surfaces

likely causes this behavior. Interkernel spread in the shortwave water vapor kernel is significantly larger near the poles, which may be due to differences in the radiative characteristics of the surface (e.g., sea ice extent, snow cover, etc.) in the kernel base states.

The surface temperature and albedo kernels are 3-dimensional, so the annual- and zonal averages only vary with latitude (Fig. 1h-k). The surface temperature kernel is highly sensitive to clouds, especially in the extratropics, as evidenced by the

325 presence of local maxima near 60°N/S in the clear-sky kernels only (Fig. 1h-i). Interkernel spread is relatively constant with latitude. The surface albedo kernel is also sensitive to clouds, especially near 60°S (Fig. 1j-k). Interkernel spread in the all-sky surface albedo kernels is largest in the tropics and largely constant with latitude in the clear-sky kernels. In the next section, we explore how the spatial structure of the kernel means and interkernel spread influence the resulting radiative feedbacks.

### 4.1.1 Feedback results

Having quantified the kernels themselves, we now focus on how the interkernel differences translate into differences in individual feedbacks. Recall that all feedbacks are calculated using identical methodologies between kernels and with the same CESM1-LE sample model output data described in Section 2.2, so all feedback variability is solely the result of the kernel choice. All feedbacks were computed using the difference in the monthly climatology of the last 30 years of the preindustrial control and a standard 150-year-long $2\times CO_2$ simulation.

Fig. 2 shows the kernel mean (left column) and standard deviation (right column) of the lapse rate, Planck, water vapor, surface albedo, and cloud feedbacks, vertically integrated when required. Generally, the lapse rate and Planck feedbacks' mean and standard deviation are of greater magnitudes at the poles (Fig. 2a-d). The strong latitudinal gradient and sign change in the lapse rate feedback (Fig. 2a) are well-recognized features in climate model simulations subjected to increasing $CO_2$. They are products of latitudinal differences in lower- and upper-tropospheric coupling, sea ice loss, and heat transport (Manabe and

Wetherald, 1975; Graversen et al., 2014; Feldl et al., 2020; Colman and Soden, 2021; Previdi et al., 2021). Similarly, the kernel mean Planck feedback is most negative over the Arctic (Fig. 2c), where the surface temperature increase is greatest via Arctic amplification, leading to the large increases in outgoing longwave radiation via the Stefan-Boltzmann law. Another area of strong negative Planck feedbacks occurs over the Southern Ocean (Fig. 2c). The overlapping maxima in interkernel spread in the lapse rate and Plank feedbacks over the Arctic and Southern Ocean indicate that these regions have the greatest sensitivity

to kernel choice (Fig. 2b,d).

The water vapor feedback is most positive in the tropical Pacific, where the increase in water vapor concentration per degree of warming is greatest, following the Clausius-Clayperon relationship (Fig. 2e). Interkernel spread is also maximized in the tropics





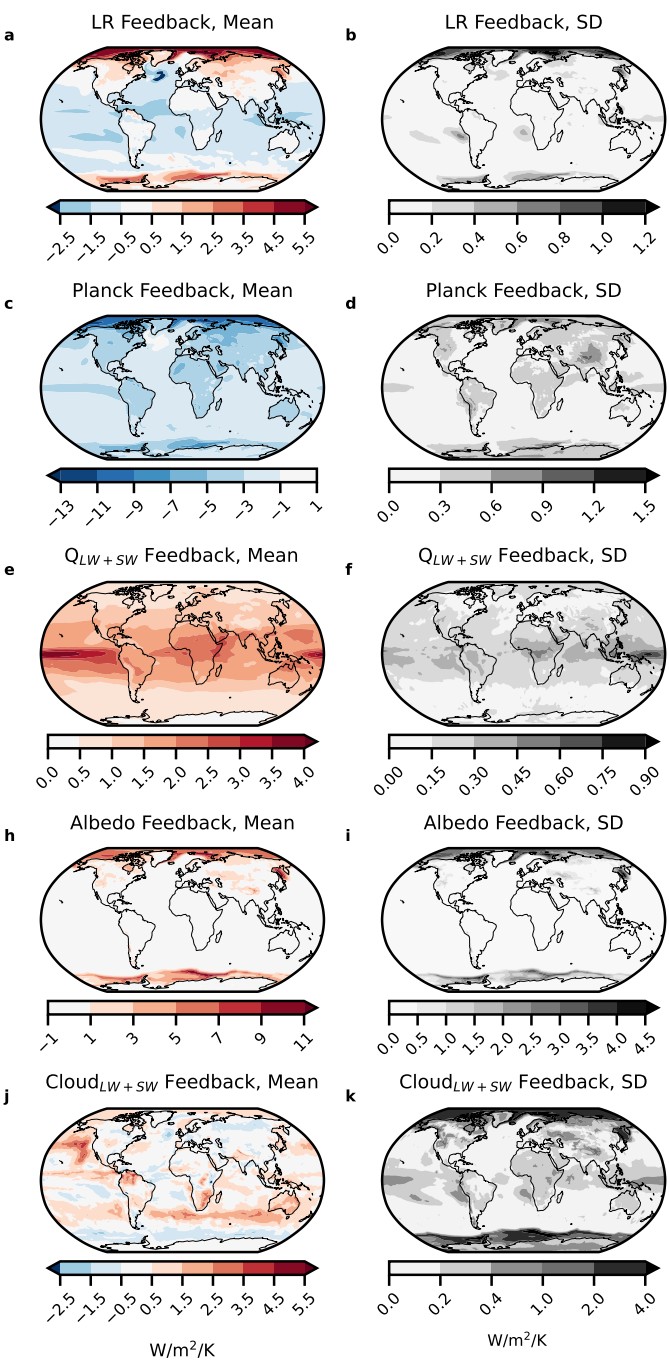

**Figure 2.** Left column: annual mean, kernel-mean (a) lapse rate (c) Planck (e) water vapor (h) surface albedo and (j) cloud feedbacks calculated using the adjustment method ($Wm^{-2}K^{-1}$). Right column: as for the left column, but showing twice the standard deviation (SD) among kernels. Feedbacks are expressed in units of $Wm^{-2}K^{-1}$ by normalizing by the global mean surface temperature response. Note the different color bar scales between feedbacks.





but exhibits a markedly different spatial distribution, with the greatest variability located over the Western Pacific (Fig. 2f). The maximum in standard deviation in the Western Pacific extends along the Equator and to the southeast, suggesting that this feature may be related to the double Intertropical Convergence Zone (ITCZ) bias present in many climate models (Lin, 2007; Tian and Dong, 2020). Additionally, there is a local standard deviation maximum in the South Atlantic Convergence Zone off the southeastern coast of South America. These maxima suggest large differences in the base states within the tropics across the eleven water vapor kernels.

The surface albedo feedback is largest over high latitude oceans (Fig. 2h-i), driven by sea ice loss (Curry et al., 1995; Riihelä et al., 2021). This sea ice loss leads to large bottom-heavy warming in these regions, resulting in a strong positive lapse rate feedback, negative Planck feedback, and similarity in the spatial patterns of the lapse rate, Planck, and surface albedo feedbacks (Croll, 1875; Ingram et al., 1989; Previdi et al., 2021). It is important to note that in the Arctic and Southern Ocean, the interkernel spread in the surface albedo feedback is larger than the spread of the lapse rate, Planck, and water vapor feedbacks, with the albedo feedback standard deviation nearly as much as 50% of the interkernel mean. Considering that there is little difference in the spread between the all-sky vs. the clear-sky albedo kernels (Fig. 1j-k), it is likely not the clouds but the base-state sea ice conditions that produce the polar-amplified spread in albedo feedback.

The last feedback we consider is the total cloud feedback in Fig. 2j-k. The kernel mean cloud feedback shows considerable spatial inhomogeneity, tending to be more negative over the oceans and positive over land, and has maxima in Equatorial South America and Africa (Fig. 2j). However, the interkernel spread in the cloud feedback is maximized over the Arctic and Southern Ocean (Fig. 2k). We show below that this is mainly due to the spread of shortwave cloud feedbacks.

Having analyzed the spatial distribution of interkernel spread, we focus on the differences between individual kernels in the zonal mean feedbacks in Fig. 3. The lapse rate and Planck feedbacks show minimal spread throughout the tropics and midlatitudes, with the greatest spread in the Arctic (Fig. 3a-b). The lapse rate feedback varies between 4 and 6 $\mathrm{Wm}^{-2}\mathrm{K}^{-1}$ poleward of 80°N but varies by less than 1 $\mathrm{Wm}^{-2}\mathrm{K}^{-1}$ elsewhere (Fig. 3a). For the Planck feedback, interkernel spread is generally less than 1 $\mathrm{Wm}^{-2}\mathrm{K}^{-1}$ everywhere and uniform in the zonal mean (Fig. 3b). The water vapor feedback is most sensitive to kernel choice in the tropics with a spread of 0.5 $\mathrm{Wm}^{-2}\mathrm{K}^{-1}$ (Fig. 3c), similar to the spatial map (Fig. 2f).

We find very large interkernel spread in the surface albedo feedback (Fig. 3d). The zonal mean Arctic surface albedo feedback spread is comparable to the kernel mean feedback value. The Southern Ocean shows a similar but weaker interkernel surface albedo feedback spread. These features are of particular importance in polar amplification studies, which we discuss in Section 5.

The cloud feedbacks are similarly sensitive to kernel choice because they are prone to uncertainties in the other feedback and radiative forcing terms, even when using the adjustment method (Soden et al., 2008). The interkernel variation in the longwave cloud feedback is largest at the poles, so even its sign in the Arctic depends on kernel choice (Fig. 3e). The shortwave cloud feedback (Fig. 3f) is likewise most sensitive to kernel choice in the high latitudes, with a zonal distribution of variability similar

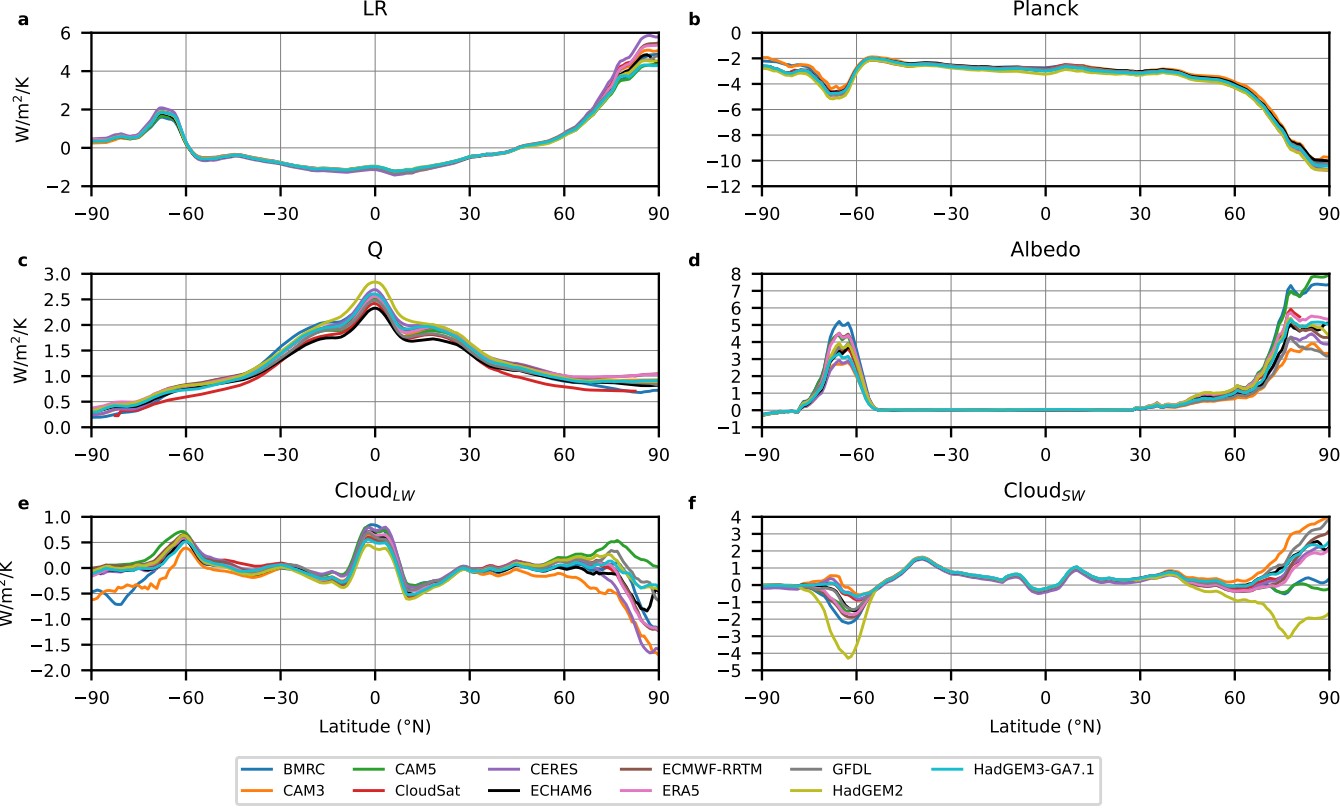

**Figure 3.** Annual zonal mean (a) lapse rate, (b) Planck, (c) total water vapor, (d) surface albedo, (e) longwave cloud, and (f) shortwave cloud feedbacks calculated using the adjustment method for each of the eleven kernels included in ClimKern.

to that of the surface albedo feedback. This is likely not a coincidence: the surface albedo feedback is used to calculate the shortwave cloud feedback via the adjustment method, so a large spread in the former translates to a large spread in the latter.

We include the global average feedback values for all 11 kernels in Table 2, along with the multikernel mean and standard deviation: note the significant interkernel spread in the albedo and cloud feedbacks relative to their mean values. Our results, therefore, demonstrate the large sensitivity of feedback to kernel choice, a factor often overlooked by many climate feedback studies.

## 5 Conclusions

The radiative kernel method is a popular and efficient way of diagnosing radiative feedbacks in climate model simulations. We were motivated to develop ClimKern to streamline these sometimes complicated calculations, shed light on kernel choice's importance in feedback studies, and provide access to a growing collection of existing kernels. We used ClimKern to compute




**Table 2.** The global annual mean feedback values (in $\mathrm{Wm^{-2}K^{-1}}$) calculated using each kernel and the same sample CESM1-LE data; from left to right, they are the lapse rate feedback, Planck feedback, total (longwave + shortwave) water vapor feedback, surface albedo feedback, and the shortwave, longwave, and total cloud feedbacks calculated using the adjustment method. The last two rows contain the kernel mean and standard deviation of the feedbacks.

| kernel | Lapse Rate | Planck | $Q_{total}$ | Albedo | $Cloud_{SW,adj}$ | $Cloud_{LW,adj}$ | $Cloud_{total,adj}$ |
|---|---|---|---|---|---|---|---|
| BMRC | -0.41 | -3.07 | 1.52 | 0.57 | 0.31 | 0.00 | 0.31 |
| CAM3 | -0.40 | -2.99 | 1.48 | 0.32 | 0.51 | -0.07 | 0.44 |
| CAM5 | -0.43 | -3.16 | 1.48 | 0.54 | 0.28 | 0.10 | 0.38 |
| CERES | -0.42 | -3.14 | 1.54 | 0.36 | 0.27 | 0.06 | 0.33 |
| CloudSat | -0.42 | -3.02 | 1.34 | 0.43 | 0.39 | 0.02 | 0.40 |
| ECHAM6 | -0.39 | -3.07 | 1.37 | 0.41 | 0.38 | -0.01 | 0.37 |
| ECMWF-RRTM | -0.38 | -3.21 | 1.53 | 0.51 | 0.30 | -0.00 | 0.30 |
| ERA5 | -0.37 | -3.18 | 1.51 | 0.52 | 0.33 | 0.01 | 0.34 |
| GFDL | -0.41 | -3.12 | 1.44 | 0.38 | 0.38 | 0.03 | 0.42 |
| HadGEM2 | -0.39 | -3.35 | 1.59 | 0.49 | 0.06 | -0.03 | 0.03 |
| HadGEM3-GA7.1 | -0.39 | -3.17 | 1.50 | 0.41 | 0.44 | -0.01 | 0.43 |
| **mean** | **-0.40** | **-3.13** | **1.48** | **0.45** | **0.33** | **0.01** | **0.34** |
| **std** | **0.02** | **0.09** | **0.07** | **0.08** | **0.11** | **0.04** | **0.11** |

basic radiative feedbacks from a sample climate model output to quantify kernel differences, leading us to the following conclusions.

**ClimKern makes radiative feedback calculations with kernels considerably easier while standardizing the underlying assumptions and methods.** The ClimKern python package contains straightforward, one-line commands for the most common calculations required for computing radiative feedbacks and can automatically load in data from the ClimKern data repository.

The code is well-documented and easily accessible on GitHub and the Python Package Index for full transparency. Operations like vertical integration or horizontal regridding are consistent across the functions, even while offering the user different options. The repository similarly employs a standard and consistent nomenclature across all kernels, making it a practical resource for anyone wishing to compute radiative feedbacks.

**Kernel choice is a nonnegligible source of uncertainty in radiative feedback calculations, especially in the polar regions.**
In terms of global average feedbacks, the lapse rate and Planck feedbacks appear to be the least sensitive to kernel choice. In contrast, the surface albedo and cloud feedbacks show considerably more sensitivity to the choice of kernel (Table 2). Interkernel spread is horizontally and vertically inhomogeneous, with all but the water vapor feedback showing greatest kernel sensitivity at the poles (Fig. 2); this may be a result of either differences in the base states or radiative schemes used to produce



the radiative kernels. In the case of the surface albedo kernels, we note that the spread is not appreciably different between the all-sky and clear-sky versions, suggesting that clouds are not the dominant cause of the spread across kernels.

Polar amplification studies frequently use the radiative kernel method to compare surface warming contributions at the poles to the global or tropical average, and then rank the relative importance of the individual feedbacks (Pithan and Mauritsen, 2014; Stuecker et al., 2018; Previdi et al., 2020; Hahn et al., 2021; Janoski et al., 2023). From the Arctic (> 70°N) values in Fig. 3 and the global average values in Table 2, the most important feedback contributing to Arctic amplification appears to be the lapse rate or surface albedo feedback, depending on the kernel used. Therefore, kernel choice can affect the conclusions of polar amplification studies, leading to our final point.

**Future studies invoking the calculations of climate feedbacks can be more robust if they include a discussion of the sensitivity of the results to kernel choice.** One option would be to use multiple kernels from the ClimKern repository in a sensitivity analysis to explore this. Another option would be to employ a kernel mean we provide here, instead of individual kernels, to limit the influence—and potential biases—of any one kernel. This option is particularly promising, given the consistent outperformance of multi-climate-model ensemble means over individual models in various metrics (Kharin and Zwiers, 2002; Tebaldi and Knutti, 2007; Bellucci et al., 2015). Future work may include comparing the sensitivity to kernel choice to other sources of uncertainty in climate studies and evaluating kernel mean performance compared to individual kernels in the computation of radiative feedbacks.

We intend for ClimKern to become a community-wide project and invite potential collaborators to contribute. The easiest way is to visit the ClimKern GitHub and fork the repository. New features and bug fixes can also be requested there.

*Code and data availability.* The ClimKern kernel and data repository is located at https://zenodo.org/records/11165999. The version of the ClimKern Python package documented in this work can be found at https://zenodo.org/records/11002820. Those interested in contributing to ClimKern or wishing to use the latest version should instead navigate to https://github.com/tyfolino/climkern. The Jupyter notebook containing the code to produce our figures is located at https://zenodo.org/records/13314165 (Janoski, 2024).

*Author contributions.* TPJ and IM led the project development. TPJ performed the majority of the coding, conducted all data analysis, and generated the figures. TPJ also wrote most of the manuscript, with contributions from IM, who helped write Sections 1 and 2.2, co-conceptualized the project, and assisted in planning the manuscript and figures. IM also contributed to the package's development with some minor coding. RJK's earlier work provided foundational insights, and he, along with MP, offered technical support throughout the project. LMP contributed by offering broader guidance and helped refine the manuscript. All authors contributed to editing the manuscript.

*Competing interests.* We declare no competing interests.



*Acknowledgements.* TPJ is supported, in part, by a National Oceanic and Atmospheric Administration (NOAA) grant via the NOAA Center for Earth System Sciences and Remote Sensing Technologies. IM is supported by a Harry Hess post-doctoral fellowship from Princeton Geosciences. LMP and MP are supported, in part, by grants from the US National Science Foundation to Columbia University. We gratefully

acknowledge the creators of the kernels used in this work for making their kernels available and the developers of Xarray and xESMF for creating software to make this work possible. TPJ thanks the US Research Software Sustainability Institute and associated instructors for teaching him how to create a Python package.



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
