# Peer review of "ClimKern v1.2: a new Python package and kernel repository for calculating radiative feedbacks"

_EGUsphere, 2024_

## Referee Comment (RC1)

**Review of "ClimKern v1.1.2: a new Python package and kernel repository for calculating radiative feedbacks"**
by Janoski et al
MS No.: egusphere-2024-2561

**Summary**

In this paper the authors describe a new python package for computing radiative feedbacks using radiative kernels and a corresponding repository of 11 radiative kernels that have been developed by various groups since the technique was introduced in 2008. The authors have brought these kernels together, placed them on consistent grids, given them consistent sign and variable naming conventions, and done additional curation in an effort to better facilitate community usage. At this time, only a subset of the most commonly used kernels (non-cloud kernels for top-of-atmosphere radiation) are part of the repository, with future plans to incorporate other kernels that are used in the community (e.g., cloud radiative kernels and kernels for surface radiation). The python package that the authors have developed for using the kernels to compute radiative flux anomalies is a major advance, as authors wishing to compute radiative feedbacks have generally either had to write code from scratch or follow someone else's code that is generally not well documented, commented, etc. It has basically been the wild west on this front for ~15 years. Given the number of methodological choices that need to be made in computing radiative feedbacks with kernels – choices that can have sizable impacts on the resulting feedback values – it is not ideal for the community of practice to be reinventing the wheel for these calculations. Having a dedicated package to perform these calculations and to quickly assess sensitivity to kernel and some methodological choices is very much welcome. I found the paper to be well written and illustrated, and I recommend acceptance of this manuscript after the revisions detailed below.

**Major Comments**

- **Role of effective radiative forcing (ERF) in the calculations.** First, on L134, I suggest providing a little more detail here regarding how ERF is computed. This is an input for the adjusted CRE calculation, so if users wanting to compute feedbacks outside of the tutorial dataset will need to know how to compute ERF. (Side note: is it worth at some point incorporating an ERF calculation capability into ClimKern?) Second, and more importantly, what if the end-user does not have ERF or chooses not to provide it? Can the calc_cloud_LW and calc_cloud_SW functions still be used if ERF is not provided? In the case of abrupt-4xCO2 simulations, the forcing is not changing through the course of the run, so if one is computing feedbacks via regression of the TOA anomalies on global mean surface air temperature (Gregory et al. 2004), the ERF term in Eq 4 and the ERF masking term in equation 5 should be zero (I think – correct me if wrong). Alternatively, when computing feedbacks from idealized atmosphere-only warming experiments (e.g., amip-p4K minus amip), there is no radiative forcing, so this term is zero by definition. I suppose the end user could provide a DataArray of zeros for the ERF term, but this is sort of klunky relative to the code allowing for this to be an optional input field.

- **Computing water vapor anomalies (L220-234).** Feedback junkies like myself have been down this dark and lonesome road, but the average reader is likely to get rather lost in this section. I think providing the relevant equations would help the reader to understand that there is some ambiguity in the right way to compute humidity anomalies and to better rationalize the four choices. A follow up question is have you assessed whether one these four choices is clearly superior and/or whether one or more are clearly inferior? Surely they can't all be equally useful, right? I think you may be in a unique position to weigh in on this, or at least report on a null

result. In my own experimentation, I seem to recall these things tending to be equivocal – some methods work better for some models and some work better for others; do you find the same?

- **Clear-sky linearity tests to evaluate kernels.** Related to the previous comment, I was surprised that you did not present clear-sky linearity tests (Shell et al. 2008), which would allow for an evaluation of which kernels best close the TOA energy budget. There seems to be a desire not to evaluate whether certain kernels are better or worse, but this would be a very useful thing for the community to know. I suppose one issue is that you have only applied the kernels to a single model, one that happens to have a corresponding kernel, which could give it an advantage in this test. So I understand the choice not to weigh in on this. However, I can't understand the statement in the conclusions that using a mean kernel would be advantageous in computing feedbacks. If one kernel is superior, then averaging it with inferior kernels should not improve things. I would expect, for example, that kernels built in late-2000s era climate models (Soden et al. 2008) would be inferior to those built from more modern GCMs or reanalyses with vastly better mean-state cloud properties and improved representation of gas optics in the radiative transfer schemes (Huang and Huang 2023). I recommend deleting these statements in the conclusions.

**Minor Comments**
- Title: should "v1.1.2" be in the title? Most of what is described is applicable beyond this specific version, I would presume.
- L62-66: It may be worth noting that Zelinka et al. (2020) assessed sensitivity of results to kernel choice as well (their Figure S2).
- L135: Why is the IRF provided?
- L190: suggest clarifying that the tropopause height input is optional
- L243: Somewhere in here I think you need to mention that the package computes all the previously described feedbacks for clear-sky conditions as well, using the respective clear-sky radiative kernels. Otherwise when you get to the cloud feedback calculations, it is unclear where the clear-sky feedbacks come from.
- L245: should "most" be "all"?
- Eq 5: I think there should be parentheses around the two $\Delta R_i$ terms that follow the summation. Also I think the nomenclature could be confusing, since the subscript "all-sky" appears in some equations but not in others.
- L304-305: it is stated that each kernel exhibits differences in the standard deviations; I think you mean "as evidenced in the standard deviations" or something like that? Also, each kernel exhibits differences between the all- and clear-sky versions. That doesn't seem surprising to me. Or are you talking about the interkernel differences in how different the all- and clear-sky kernels are? I think this sentence needs to be re-written for clarity, since the first part deals with inter-kernel spread while the latter deals with all- vs clear-sky differences within a given kernel (I think).
- L319: I would have thought solar path length through the atmosphere would be highly relevant too.
- L330-334: I think you should provide more explicit detail about how you did your feedback calculations here. Which WV feedback option was used? Did you integrate up to the default tropopause, or did you compute the tropopause explicitly? (Side note: is it worth at some point incorporating a tropopause calculation function into ClimKern, something like PyTropD?). To clarify: are you differencing a climatology from the last 30 years of abrupt simulation and a climatology from the last 30 years of the piControl simulation, or are you using a climatology

from the full 150-year abrupt simulation? In either case, I suggest mentioning that this difference of perturbed and control climatologies is not ideal for computing feedbacks in abrupt 2x or 4x CO2 runs because rapid adjustments are aliased into the feedback when computed this way. Computing the TOA anomalies throughout the duration of the 150-year abrupt experiment and regressing them on coincident global mean surface air temperature anomalies is preferred. Related to this, does the code require that both the perturbed and control data inputs have no more than 12 months? Can one input perturbed climate fields that are length N*12 months (where N is the number of years) and have the code difference them with the 12-month long piControl climate, yielding N*12 month TOA anomalies?

- Figure 2: Why is the standard deviation multiplied by 2? I don't love how the colorbar scales change among the figures, especially for the right column. Could the standard deviation colorbars be objectively related to the means (e.g., from 0 to some percentage of the range of mean magnitudes)? Currently the tiny interkernel WV and Planck feedback spreads are over-emphasized relative to, say, the cloud feedback spread.
- Section 4.1.1: Suggest reiterating somewhere in here (or at multiple places) that these are just results from a single model (CESM1-LE). Also, I may have missed it, but are you using just one ensemble member? Are the other members of the LE just used for diagnosing ERF?
- L363: I don't really see this (much of the ocean has a positive cloud feedback and much of the land has a negative cloud feedback), so I don't think it should be the primary feature to highlight.
- Figure 3: The fact that the y-axis ranges are so different (some only span 3 W/m2/K while others span 12 W/m2/K) tends to mislead regarding interkernel spread. I think these should either be put on equal footing or this plotting choice should be pointed out more explicitly.
- L400: I don't see this. Table 2 shows that the interkernel standard deviation of Planck is larger than for 3 other feedbacks (WV, surface albedo, and LW cloud). Are you referring to the sensitivity as a fraction of the mean?

**References**

Gregory, J. M., and Coauthors, 2004: A new method for diagnosing radiative forcing and climate sensitivity. *Geophys. Res. Lett.*, **31**, https://doi.org/10.1029/2003GL018747.

Huang, H., and Y. Huang, 2023: Radiative sensitivity quantified by a new set of radiation flux kernels based on the ECMWF Reanalysis v5 (ERA5). *Earth Syst. Sci. Data*, **15**, 3001–3021, https://doi.org/10.5194/essd-15-3001-2023.

Shell, K. M., J. T. Kiehl, and C. A. Shields, 2008: Using the Radiative Kernel Technique to Calculate Climate Feedbacks in NCAR's Community Atmospheric Model. *J Clim.*, **21**, 2269–2282, https://doi.org/10.1175/2007JCLI2044.1.

Soden, B. J., I. M. Held, R. Colman, K. M. Shell, J. T. Kiehl, and C. A. Shields, 2008: Quantifying Climate Feedbacks Using Radiative Kernels. *J Clim.*, **21**, 3504–3520, https://doi.org/10.1175/2007JCLI2110.1.

Zelinka, M. D., T. A. Myers, D. T. McCoy, S. Po-Chedley, P. M. Caldwell, P. Ceppi, S. A. Klein, and K. E. Taylor, 2020: Causes of Higher Climate Sensitivity in CMIP6 Models. *Geophys. Res. Lett.*, **47**, e2019GL085782, https://doi.org/10.1029/2019GL085782.

---

## Referee Comment (RC2)

General comments:

This manuscript is a significant contribution to the field of radiative feedback and radiative adjustments studies where radiative kernels are frequently used but, as the authors note, not always readily accessible. This has the potential to advance modelling science of particularly radiative feedback studies, but also studies of radiative adjustments. The tool itself should be very useful for scientists in the field, and while the analysis presented comparing different kernels is not extensive, it is sufficient to illustrate some interesting novel results and could easily lead to further investigation. The manuscript is well written, with the methods used mostly very clear and the results clearly support the interpretations and conclusions made. A few points would however, benefit from clarification with additional information or rewording. The structure of the manuscript is clear and concise, the manuscript reads well, and the abstract and title describe the contents of the manuscript well. However, in a number of places the language describing results is vague and would benefit from the use of actual quantified values (most of which could be derived from the figures). Including values in the conclusion and abstract would really help to emphasise the key results and highlight the interesting results the authors have found to readers. Other papers are properly referenced and only one or two points are missing a reference. The supplementary material documenting the code and datasets is extensive and should be very useful to potential users. Overall the manuscript is a great contribution to the literature and I suggest only minor amendments - some to clarify the method and results, and others to maximise its usefulness to potential users - and can only wish such a tool were available when I first became interested in radiative kernels!

Specific comments:

- There are a number of instances in the paper, particularly the results, where results are described only qualitatively. Generally it would be better to give quantified values in text, which also describe the sign of the change, rather than words like 'highly' or 'considerably' which are subjective and sign agnostic. While readers can see results in the figures, using actual values in the text would help make the key results and their significance clearer to the reader. I've noted more examples in the line by line comments below, but for example in line 324, you could say that 'the clear sky kernel is up to 1 W m^-2 more negative than the all-sky kernel', rather than just 'the surface temperature kernel is highly sensitive to clouds'
- While table 1 offers some useful comparison of the differences between the kernels, there are a number of factors not included, some of which may be of greater importance. Firstly, the number of years averaged over to generate the kernel would seem as important as number of levels and resolution. Likewise, whether the kernel was computed with aerosol included or not (for example, the HadGEM3 kernel was computed without aerosol and so effectively represents a 'clean-sky' kernel; whereas the CAM5 and ERA5 kernels were computed with aerosol represented in the radiation code) could have a big effect and is worth stating. Other differences may be less important, but perhaps could still be included as columns, such as the reference climate state used (e.g. pre-industrial or present-day), and the data source used (e.g. model, reanalysis, satellite). While readers could find these details from reading the referenced papers, it would seem very useful to readers and users of the code to include a few more of these differences here.
- On lines 172-173 and 239-240, if I understand correctly, the functions calculate a monthly mean climatology from the control simulation input and subtract from the un-averaged perturbed simulation input? Is it expected that the input from the perturbed simulation will already be given as monthly means too? If not, then what is the reason for taking monthly climatology of the control experiment input, but differencing to non-monthly meaned perturbed experiment input? It may be the sentences just need re-wording.

- On line 208, what does 'masking below the surface' mean? Presumably this is related to how surface/orography is defined differently among kernels and models, but it would be good to explain this, and perhaps explain somewhere how these differences are dealt with. And state if this is also applied to the other kernels not just the water vapour kernel.
- It would be great to evidence and emphasise the second conclusion point by stating what is the largest difference between two kernels in terms of the total climate feedback, or simply the stdev of the sum of feedbacks across the kernels. This would also be a great 'headline' result to show readers in the abstract and help quantify the text in lines 13-14. (Perhaps going further, you could even express this in terms of the uncertainty it would add to the ECS calculated from the sample input, but that might add too many factors, so is just an idea).
- On lines 414-415, while a discussion around the validity of multi-model-mean approaches is much bigger than needs addressing here, there are issues with encouraging users to use the mean of all the kernels as a simple way to eliminate biases. For example, clearly some kernels are not independent, such as CAM3 and CAM5 or HadGEM2 and HadGEM3 kernels and could share similar biases. Perhaps you could re-word to suggest that using the mean of several kernels that the user selects (in addition to using multiple kernels for sensitivity analysis as suggested in your previous sentence) might be better than a single model, but the user should decide which ones to include.
- Not essential but more of a question and observation: lines 380-381, since the spread in the SW cloud feedback is influenced by the spread in the surface albedo feedback, would you expect the relative bias of each kernel to the mean to be somewhat inverted between the two feedbacks? It looks like one does see that to a degree: CAM3 and GFDL are the highest over the Arctic for SW cloud but lowest for albedo, whereas the reverse is true for BMRC and CAM5 (excluding HadGEM2, which perhaps has some other differences with the SW cloud kernel). It would be nice to mention this if you agree. And also perhaps suggest an explanation why the HadGEM2 kernel is such an outlier for SW cloud feedback
- I noticed that the ECHAM5 kernel is included in the kernel data repository alongside the 11 mentioned in the paper - is there a reason this was included in the repository data but not in the analysis in the paper?

Technical corrections

- 11 - would be great to specify here the sample climate model output is representing a 2x CO2 perturbation
- 13-14 - would be nice to include some quantification of the spread - would highlight the significance of the results to readers
- 65 - quantify how much the surface albedo kernel varied by
- 66 - again ideally quantify
- 77-79 - need reference(s) for these examples
- 146 - Smith et al 2018 may not be the first ref for this - other studies used kernels for adjustments earlier (e.g. Vial et al 2013, Block and Mauritsen 2013)
- 160 - define 'a control and a perturbed simulation' - it might not be immediately obvious to all readers
- 188 - 'holding the control and perturbed simulations' - should this be 'holding the input data of the control and perturbed simulations'?
- 241-242 - is the option to calculate all-sky or clear-sky feedbacks also applicable to other functions? If so, it would be worth mentioning it earlier in the first function which it is true for, or in the intro to section 3

- 250-252 - while its great the kernels can be used for radiative adjustments too, I am not sure the cloud kernel will work for radiative adjustments with either the adjustment or residual method equations given in section 3.4 - this may be worth noting
- 306 - strictly speaking, these are minima for the means, being negative not positive, but perhaps could be described as 'maxima in magnitude'
- 309-310 - would be good to suggest an explanation of how the inclusion of clouds or not might cause such a change in the spatial pattern of the mean temperature kernel
- Figure 1 and figure 2 both skip the subfigure label 'g' and go straight to 'h'
- 320 - avoid use of 'significantly' where not referring to statistical significance; 'larger' would suffice, or better yet quantify it
- 329 - The numbering of the section would make more sense as 4.2 here rather than 4.1.1 (as there is presently no section 4.2, nor a 4.1.2 either)
- 333-334 - explain whether using the last 30 years of a 150 year simulation means that the surface T response has equilibrated sufficiently, or use a reference to state this is a sufficiently long enough length of time to use
- 363-364 - The description of figure 2j is inaccurate: the maxima over NE Pacific and the southern Atlantic and Indian Oceans are similarly large to the two mentioned in the text, and the contrast between land and sea is not as clear as between different land areas or different sea areas. I would suggest a different description.
- Figure 2 - it would be helpful to separate the cloud feedback into LW and SW components, to match the treatment as in figure 3. This should help with the description of the spatial pattern in the text too. Perhaps this could be achieved without reducing the subfigure size (and still fitting on one page) by making it 4 columns instead of 2 columns of subfigures.
- 372 - 'very large' - please quantify (especially because it is an interesting result!)
- 404-405 - this is inconsistent with line 328, where you noted that the surface albedo kernel spread is indeed different between the all and clear sky in the tropics; I'm also not sure that this would prove the point that clouds are not the dominant cause of spread across kernels. I do agree however, in figure 1, that only the T kernel spread seems to differ a lot between all and clear sky - so maybe re-word this sentence if you want to make this point, or remove it.
- 408-410 - I am not clear how the global mean feedbacks in table 2 support the interpretation that the lapse rate or surface albedo feedbacks are most important for Arctic amplification?

---

## Author Comment (AC1)

**Reviewer # 1**

Review of "ClimKern v1.1.2: a new Python package and kernel repository for calculating radiative feedbacks" by Janoski et al MS No.: egusphere-2024-2561

**Summary** In this paper the authors describe a new python package for computing radiative feedbacks using radiative kernels and a corresponding repository of 11 radiative kernels that have been developed by various groups since the technique was introduced in 2008. The authors have brought these kernels together, placed them on consistent grids, given them consistent sign and variable naming conventions, and done additional curation in an eAort to better facilitate community usage. At this time, only a subset of the most commonly used kernels (non-cloud kernels for top-of-atmosphere radiation) are part of the repository, with future plans to incorporate other kernels that are used in the community (e.g., cloud radiative kernels and kernels for surface radiation). The python package that the authors have developed for using the kernels to compute radiative flux anomalies is a major advance, as authors wishing to compute radiative feedbacks have generally either had to write code from scratch or follow someone else's code that is generally not well documented, commented, etc. It has basically been the wild west on this front for ~15 years. Given the number of methodological choices that need to be made in computing radiative feedbacks with kernels – choices that can have sizable impacts on the resulting feedback values – it is not ideal for the community of practice to be reinventing the wheel for these calculations. Having a dedicated package to perform these calculations and to quickly assess sensitivity to kernel and some methodological choices is very much welcome. I found the paper to be well written and illustrated, and I recommend acceptance of this manuscript after the revisions detailed below.

Major Comments

• Role of effective radiative forcing (ERF) in the calculations. First, on L134, I suggest providing a little more detail here regarding how ERF is computed. This is an input for the adjusted CRE calculation, so if users wanting to compute feedbacks outside of the tutorial dataset will need to know how to compute ERF. (Side note: is it worth at some point incorporating an ERF calculation capability into ClimKern?) Second, and more importantly, what if the end-user does not have ERF or chooses not to provide it? Can the calc_cloud_LW and calc_cloud_SW functions still be used if ERF is not provided? In the case of abrupt-4xCO2 simulations, the forcing is not changing through the course of the run, so if one is computing feedbacks via regression of the TOA anomalies on global mean surface air temperature (Gregory et al. 2004), the ERF term in Eq 4 and the ERF masking term in equation 5 should be zero (I think – correct me if wrong). Alternatively, when computing feedbacks from idealized atmosphere-only warming experiments (e.g., amip-p4K minus amip), there is no radiative forcing, so this term is zero by definition. I suppose the end user could provide a DataArray of zeros for the ERF term, but this is sort of klunky relative to the code allowing for this to be an optional input field.

We estimate the ERF with abrupt 2xCO2 and 1xCO2 experiments, as per Forster et al. (2016), with prescribed SSTs and sea-ice concentrations at piControl levels. These experiments are 30 years long; we take the average over the last 10 years. We calculate ERF as the difference between the global mean net TOA flux between PI and 2xCO2 in these prescribed SSTs and SICs experiments. We do not adjust for land warming. We have revised the text to reflect this. (L134-138)

You are correct that if you calculate the radiative feedbacks using the regression method, the radiative forcing term is a constant with no impact on the resulting feedback. We have changed all four functions for calculating the LW and SW cloud feedbacks such that the radiative forcing terms are optional and, if not provided by the user, are assumed to be 0. We also added this information in L264-266.

ClimKern may include a function to calculate radiative forcings in a future iteration, but we are hesitant to include such a feature given the complexities it would introduce.

• Computing water vapor anomalies (L220-234). Feedback junkies like myself have been down this dark and lonesome road, but the average reader is likely to get rather lost in this section. I think providing the relevant equations would help the reader to understand that there is some ambiguity in the right way to compute humidity anomalies and to better rationalize the four choices. A follow up question is have you assessed whether one these four choices is clearly superior and/or whether one or more are clearly inferior? Surely they can't all be equally useful, right? I think you may be in a unique position to weigh in on this, or at least report on a null result. In my own experimentation, I seem to recall these things tending to be equivocal – some methods work better for some models and some work better for others; do you find the same?

Excellent point—it is easy to overestimate how much experience the reader has with these esoteric techniques! We have made considerable changes to Section 3.2 by adding more complete descriptions and equations. Please see our new L220-235.

Regarding the different water vapor feedback algorithms, we added Table S1, which shows the global, annual mean water vapor feedbacks calculated with methods 1-4, and corresponding discussion in L416-422. We comment further on kernel rankings below.

• Clear-sky linearity tests to evaluate kernels. Related to the previous comment, I was surprised that you did not present clear-sky linearity tests (Shell et al. 2008), which would allow for an evaluation of which kernels best close the TOA energy budget. There seems to be a desire not to evaluate whether certain kernels are better or worse, but this would be a very useful thing for the community to know. I suppose one issue is that you have only applied the kernels to a single model, one that happens to have a corresponding kernel, which could give it an advantage in this test. So I understand the choice not to weigh in on this. However, I can't understand the statement in the conclusions that using a mean kernel would be advantageous in computing feedbacks. If one kernel is superior, then averaging it with inferior kernels should not improve things. I would expect, for example, that kernels built in late-2000s era climate models (Soden et

al. 2008) would be inferior to those built from more modern GCMs or reanalyses with vastly better mean-state cloud properties and improved representation of gas optics in the radiative transfer schemes (Huang and Huang 2023). I recommend deleting these statements in the conclusions.

We now include clear-sky residuals in Table S2 and L423-440. As you've pointed out, it is difficult to make generalizations using one CESM1 simulation, and we do not wish for our results to be overinterpreted. It is likely that there is not a single "best" kernel for all models and forcing scenarios, and Table S2 shows that the clear-sky residuals are dependent on the water vapor feedback algorithm.

In the absence of a "best" kernel, using a kernel mean may offer an advantage in consistency and reducing sensitivity to individual kernel biases. We have altered this section (L473-476) to be much more careful and deliberate in our wording. Our upcoming manuscript will hopefully shed more light on this subject by incorporating a multi-model ensemble.

Minor Comments

• Title: should "v1.1.2" be in the title? Most of what is described is applicable beyond this specific version, I would presume.

GMD requires version numbers in the titles of software papers, but we have updated it to version 1.2 after incorporating reviewers' suggestions in a new version of ClimKern.

• L62-66: It may be worth noting that Zelinka et al. (2020) assessed sensitivity of results to kernel choice as well (their Figure S2).

Thank you for pointing out that Zelinka et al. (2020) assessed sensitivity due to 6 kernels. We have revised the text to reflect this (L63).

• L135: Why is the IRF provided?

We plan on increasing functionality to include radiative adjustments that would require the IRF, but we have removed the lines because we have not implemented that yet.

• L190: suggest clarifying that the tropopause height input is optional

Done. (L182)

• L243: Somewhere in here I think you need to mention that the package computes all the previously described feedbacks for clear-sky conditions as well, using the respective clear-sky radiative kernels. Otherwise when you get to the cloud feedback calculations, it is unclear where the clear-sky feedbacks come from.

Thanks for the suggestion. We added a sentence at the beginning of the section (L146-147).

• L245: should "most" be "all"?

Definitely. We removed "most."

• Eq 5: I think there should be parentheses around the two ∆Ri terms that follow the summation. Also I think the nomenclature could be confusing, since the subscript "all-sky" appears in some equations but not in others.

You're right. We have added the parentheses and switched to $^{o}$ to signify clear-sky and lack thereof all-sky. It is now Eq. 10.

• L304-305: it is stated that each kernel exhibits differences in the standard deviations; I think you mean "as evidenced in the standard deviations" or something like that? Also, each kernel exhibits differences between the all- and clear-sky versions. That doesn't seem surprising to me. Or are you talking about the interkernel differences in how differences the all- and clear-sky kernels are? I think this sentence needs to be re-written for clarity, since the first part deals with inter-kernel spread while the latter deals with all- vs clear-sky differences within a given kernel (I think).

Looking back, this sentence is confusing and unnecessary. We removed it from the text.

• L319: I would have thought solar path length through the atmosphere would be highly relevant too.

It's possible, but Huang and Huang's (2023) surface SW WV kernel shows the opposite trend with greater values at the equator due to background water vapor concentrations. We haven't found a primary source directly connecting solar path length to SW WV kernels.

• L330-334: I think you should provide more explicit detail about how you did your feedback calculations here. Which WV feedback option was used? Did you integrate up to the default tropopause, or did you compute the tropopause explicitly? (Side note: is it worth at some point incorporating a tropopause calculation function into ClimKern, something like PyTropD?). To clarify: are you differencing a climatology from the last 30 years of abrupt simulation and a climatology from the last 30 years of the piControl simulation, or are you using a climatology from the full 150-year abrupt simulation? In either case, I suggest mentioning that this difference of perturbed and control climatologies is not ideal for computing feedbacks in abrupt 2x or 4x CO2 runs because rapid adjustments are aliased into the feedback when computed this way. Computing the TOA anomalies throughout the duration of the 150-year abrupt experiment and regressing them on coincident global mean surface air temperature anomalies is preferred. Related to this, does the code require that both the perturbed and control data inputs have no more than 12 months? Can one input perturbed climate fields that are length N*12 months

(where N is the number of years) and have the code difference them with the 12- month long piControl climate, yielding N*12 month TOA anomalies?

We've added the suggested details to the paragraph (L346-350). Thanks for the suggestion!

Regarding the tropopause calculation function, that is a great idea. We opened it as an issue on GitHub and plan on implementing it in the next version.

You are correct that we are defining the response as the difference between the abrupt2xCO2 and piControl simulations, and this inadvertently includes rapid adjustments in resulting feedback values. We felt this simplification was acceptable for a few reasons. First, we were able to precalculate the climatologies of the last 30 years of the abrupt2xCO2 and piControl simulations so that the sample data distributed with ClimKern is as small as possible. Having 1800 months vs. 12 months would substantially increase the data download size. Second, the main scientific question we intended to answer with this work is whether kernel choice is an important consideration for climate sensitivity studies while highlighting ClimKern. Therefore, the actual feedback values computed should be secondary to kernel spread. Third, while we recognize it is not ideal, this type of analysis is still commonly found in the literature, such as in Goosse et al. (2018), Hahn et al. (2020), and Previdi et al. (2020). We added a discussion to highlight this caveat (L352-356).

ClimKern is built to handle any amount of monthly data but will kick back an error if the number of months is not divisible by 12. Your proposed situation will work fine with ClimKern, and you can extend it further (e.g., feed ClimKern a full 1000-year piControl simulation and a 150-year abrupt2xCO2 simulation). At that point, you can regress against the temperature change to do the more Gregory-style feedback approach. We will do this in our upcoming work. We also added L147-148 to clarify that ClimKern accepts monthly mean input.

• Figure 2: Why is the standard deviation multiplied by 2? I don't love how the colorbar scales change among the figures, especially for the right column. Could the standard deviation colorbars be objectively related to the means (e.g., from 0 to some percentage of the range of mean magnitudes)? Currently the tiny interkernel WV and Planck feedback spreads are overemphasized relative to, say, the cloud feedback spread.

We have altered Figure 2 so that the means and standard deviations are on the same nonlinear color scale. Thank you for this suggestion!

• Section 4.1.1: Suggest reiterating somewhere in here (or at multiple places) that these are just results from a single model (CESM1-LE). Also, I may have missed it, but are you using just one ensemble member? Are the other members of the LE just used for diagnosing ERF?

We have revised the paragraph to emphasize that we use a single model run (L350-351).

Please note that we are using an abrupt 2xCO2 simulation from CESM1-LE to diagnose the feedbacks. We are not using the large ensemble historical and RCP runs from Kay et al. (2015), but we simply use the same model to perform our abrupt 2xCO2 run.

We run additional atmosphere-only abrupt 2xCO2 experiments with prescribed SST and SIC to piControl values to diagnose the ERF (see previous comment) in the standard way as in Forster et al. 2016.

• L363: I don't really see this (much of the ocean has a positive cloud feedback and much of the land has a negative cloud feedback), so I don't think it should be the primary feature to highlight.

You're right! We removed that statement.

• Figure 3: The fact that the y-axis ranges are so different (some only span 3 W/m2/K while others span 12 W/m2/K) tends to mislead regarding interkernel spread. I think these should either be put on equal footing or this plotting choice should be pointed out more explicitly.

Another fair critique. To address this, we altered Figure 3 so that all y-axes are the same, making the spreads comparable between feedbacks. However, we had to add constant values to the Planck and surface albedo feedbacks to achieve this.

• L400: I don't see this. Table 2 shows that the interkernel standard deviation of Planck is larger than for 3 other feedbacks (WV, surface albedo, and LW cloud). Are you referring to the sensitivity as a fraction of the mean?

Thanks for catching this. We have clarified that the Planck and surface albedo feedbacks show the greatest interkernel variability (L467-468).

References

Gregory, J. M., and Coauthors, 2004: A new method for diagnosing radiative forcing and climate sensitivity. Geophys. Res. Lett., 31, https://doi.org/10.1029/2003GL018747. Huang, H., and Y.

Huang, 2023: Radiative sensitivity quantified by a new set of radiation flux kernels based on the ECMWF Reanalysis v5 (ERA5). Earth Syst. Sci. Data, 15, 3001–3021, https://doi.org/10.5194/essd-15-3001-2023.

Shell, K. M., J. T. Kiehl, and C. A. Shields, 2008: Using the Radiative Kernel Technique to Calculate Climate Feedbacks in NCAR's Community Atmospheric Model. J Clim., 21, 2269–2282, https://doi.org/10.1175/2007JCLI2044.1.

Soden, B. J., I. M. Held, R. Colman, K. M. Shell, J. T. Kiehl, and C. A. Shields, 2008: Quantifying Climate Feedbacks Using Radiative Kernels. J Clim., 21, 3504–3520, https://doi.org/10.1175/2007JCLI2110.1.

Zelinka, M. D., T. A. Myers, D. T. McCoy, S. Po-Chedley, P. M. Caldwell, P. Ceppi, S. A. Klein, and K. E. Taylor, 2020: Causes of Higher Climate Sensitivity in CMIP6 Models. Geophys. Res. Lett., 47, e2019GL085782, https://doi.org/10.1029/2019GL085782

**Reviewer # 2**

General comments: This manuscript is a significant contribution to the field of radiative feedback and radiative adjustments studies where radiative kernels are frequently used but, as the authors note, not always readily accessible. This has the potential to advance modelling science of particularly radiative feedback studies, but also studies of radiative adjustments. The tool itself should be very useful for scientists in the field, and while the analysis presented comparing different kernels is not extensive, it is sufficient to illustrate some interesting novel results and could easily lead to further investigation. The manuscript is well written, with the methods used mostly very clear and the results clearly support the interpretations and conclusions made. A few points would however, benefit from clarification with additional information or rewording. The structure of the manuscript is clear and concise, the manuscript reads well, and the abstract and title describe the contents of the manuscript well. However, in a number of places the language describing results is vague and would benefit from the use of actual quantified values (most of which could be derived from the figures). Including values in the conclusion and abstract would really help to emphasise the key results and highlight the interesting results the authors have found to readers. Other papers are properly referenced and only one or two points are missing a reference. The supplementary material documenting the code and datasets is extensive and should be very useful to potential users. Overall the manuscript is a great contribution to the literature and I suggest only minor amendments - some to clarify the method and results, and others to maximise its usefulness to potential users - and can only wish such a tool were available when I first became interested in radiative kernels!

**Specific comments:**

• There are a number of instances in the paper, particularly the results, where results are described only qualitatively. Generally it would be better to give quantified values in text, which also describe the sign of the change, rather than words like 'highly' or 'considerably' which are subjective and sign agnostic. While readers can see results in the figures, using actual values in the text would help make the key results and their significance clearer to the reader. I've noted more examples in the line by line comments below, but for example in line 324, you could say that 'the clear sky kernel is up to 1 W m^-2 more negative than the all-sky kernel', rather than just 'the surface temperature kernel is highly sensitive to clouds'

Thank you for this feedback! We have added many quantitative statements/results to the text, including the line-by-line ones you suggested below.

• While table 1 offers some useful comparison of the differences between the kernels, there are a number of factors not included, some of which may be of greater importance. Firstly, the number of years averaged over to generate the kernel would seem as important as number of levels and resolution. Likewise, whether the kernel was computed with aerosol included or not (for example, the HadGEM3 kernel was computed without aerosol and so effectively represents a 'clean-sky' kernel; whereas the CAM5 and ERA5 kernels were computed with aerosol represented in the radiation code) could have a big effect and is worth stating. Other differences may be less important, but perhaps could still be included as columns, such as the reference climate state used (e.g. pre-industrial or present-day), and the data source used (e.g. model, reanalysis, satellite). While readers could find these details from reading the referenced papers, it would seem very useful to readers and users of the code to include a few more of these differences here.

You are correct that there are a multitude of factors that can potentially be important in explaining kernel differences. We have included a new column in Table 1 to indicate the data source (climate model, reanalysis, or satellite). However, we are uncomfortable adding more details and prefer to refer the readers to the primary sources for two reasons.

First, the documentation for each kernel ranges from complete to nearly nonexistent. For example, in efforts to address this comment, we have encountered several roadblocks trying to track down even basic details regarding some kernels', such as the $CO_2$ levels or forcing scenario in which they were generated. The language can be ambiguous even when details are present in the primary source. We do not feel comfortable with the thought of possibly misrepresenting others' work and want to avoid Table 1 becoming the authoritative source on kernel documentation.

Second, the impacts of kernel-design choices are relatively unexplored/unquantified. For example, it is unclear what differences arise from a kernel generated from a present-day simulation vs. that of a historical simulation. By adding those details to Table 1, we risk the reader overinterpreting the information when the impacts of those factors would better be explored in a separate, more thoughtfully designed study dedicated to that topic.

• On lines 172-173 and 239-240, if I understand correctly, the functions calculate a monthly mean climatology from the control simulation input and subtract from the un-averaged perturbed simulation input? Is it expected that the input from the perturbed simulation will already be given as monthly means too? If not, then what is the reason for taking monthly climatology of the control experiment input, but differencing to non-monthly meaned perturbed experiment input? It may be the sentences just need re-wording.

Your interpretation is correct. We have added a line in Section 3 to clarify that ClimKern only accepts monthly mean fields (L147-148).

• On line 208, what does 'masking below the surface' mean? Presumably this is related to how surface/orography is defined differently among kernels and models, but it would be good to

explain this, and perhaps explain somewhere how these differences are dealt with. And state if this is also applied to the other kernels not just the water vapour kernel.

Great point. We added a line in Section 3.1 and reworded a line in Section 3.2 stating that values are masked below the climatological surface pressure in the control simulation, which requires user input. (L177-178, 216-217).

• It would be great to evidence and emphasise the second conclusion point by stating what is the largest difference between two kernels in terms of the total climate feedback, or simply the stdev of the sum of feedbacks across the kernels. This would also be a great 'headline' result to show readers in the abstract and help quantify the text in lines 13-14. (Perhaps going further, you could even express this in terms of the uncertainty it would add to the ECS calculated from the sample input, but that might add too many factors, so is just an idea).

After considering this thoughtful suggestion, we decided against including a kernel-derived net climate feedback parameter. The net climate feedback parameter (λ) and ECS are the same regardless of the kernel because they are characteristics of the model simulation. Typically, the difference between the sum of kernel-derived feedbacks and the model-derived net climate feedback parameter is quantified as the residual, which we added to the manuscript with Table S2 and L423-440. However, we also do not wish to overstate the robustness of our results, given that we are only using one model simulation. We have an upcoming manuscript that will use a more representative sample of climate model simulations.

• On lines 414-415, while a discussion around the validity of multi-model-mean approaches is much bigger than needs addressing here, there are issues with encouraging users to use the mean of all the kernels as a simple way to eliminate biases. For example, clearly some kernels are not independent, such as CAM3 and CAM5 or HadGEM2 and HadGEM3 kernels and could share similar biases. Perhaps you could re-word to suggest that using the mean of several kernels that the user selects (in addition to using multiple kernels for sensitivity analysis as suggested in your previous sentence) might be better than a single model, but the user should decide which ones to include.

We have taken a more cautious approach to this section and clarified that the kernel mean may be useful in studies involving more than one climate model but noting that the "best" kernel is model-dependent (L474-476).

• Not essential but more of a question and observation: lines 380-381, since the spread in the SW cloud feedback is influenced by the spread in the surface albedo feedback, would you expect the relative bias of each kernel to the mean to be somewhat inverted between the two feedbacks? It looks like one does see that to a degree: CAM3 and GFDL are the highest over the Arctic for SW cloud but lowest for albedo, whereas the reverse is true for BMRC and CAM5 (excluding HadGEM2, which perhaps has some other differences with the SW cloud kernel). It would be nice to mention this if you agree. And also perhaps suggest an explanation why the HadGEM2 kernel is such an outlier for SW cloud feedback

This is an interesting question! It is not as straightforward as it might seem because we use the adjustment method for cloud feedbacks. Thus, the term included in the SW cloud feedback calculation is the *difference* between the clear-sky and all-sky surface albedo feedbacks, which is a positive quantity. There is little interkernel spread in the global annual mean clear-sky surface albedo feedback (below). Therefore, the kernels that produce the largest all-sky surface albedo feedback values tend to have the smallest differences from the clear-sky feedback, creating the inverse relationship you have noticed.

| kernel | $\lambda\_\alpha\_as$ | $\lambda\_\alpha\_cs$ | $\lambda\_\alpha\_cs - \lambda\_\alpha\_as$ |
|---|---|---|---|
| BMRC | 0.57 | 0.81 | 0.24 |
| CAM3 | 0.32 | 0.78 | 0.46 |
| CAM5 | 0.54 | 0.80 | 0.26 |
| CloudSat | 0.43 | 0.76 | 0.33 |
| CERES | 0.36 | 0.74 | 0.38 |
| ECHAM6 | 0.41 | 0.76 | 0.35 |
| ECMWF-RRTM | 0.51 | 0.78 | 0.26 |
| ERA5 | 0.52 | 0.79 | 0.26 |
| GFDL | 0.38 | 0.75 | 0.36 |
| HadGEM2 | 0.49 | 0.81 | 0.33 |
| HadGEM3-GA7.1 | 0.41 | 0.80 | 0.38 |

Thank you for pointing out HadGEM2. After digging, it seems we accidentally used the all-sky surface albedo kernel for the clear-sky version. We have fixed this in the repository and updated our tables and figures to reflect this.

• I noticed that the ECHAM5 kernel is included in the kernel data repository alongside the 11 mentioned in the paper - is there a reason this was included in the repository data but not in the analysis in the paper?

Good eye. Yes, the ClimKern repository contains the ECHAM5 kernel, which was created by one author of this study. We had initially intended to include this kernel set, but the resulting global average Planck (-2.73 W/m$^2$/K) and water vapor feedbacks (1.91 W/m$^2$/K) are well outside of the range from other kernels and estimates in the literature. Also, the surface albedo kernel has an incorrect seasonality. It is unclear when exactly these errors happened. We decided to exclude it from the analysis. We left it in the repository but will likely remove it in a future update.

**Technical corrections**

• 11 - would be great to specify here the sample climate model output is representing a 2x CO2 perturbation

Thanks for suggesting this. We have revised the sentence to specify that we use a 2xCO2 experiment (L11).

• 13-14 - would be nice to include some quantification of the spread - would highlight the significance of the results to readers

We wish to avoid overstating our results given that we only use one model simulation, but we added L14-15 to provide context to the relative variability of the feedbacks. Our future work will be able to more confidently provide numeric values for more than one model.

• 65 - quantify how much the surface albedo kernel varied by

Done. Thank you for the suggestion! (L66)

• 66 - again ideally quantify

It's based on Hahn et al. (2021) Fig. 5, but to my knowledge, the study does not present the numeric values. We've changed this to avoid subjectivity by noting that the relative importance of feedbacks as polar amplification mechanisms show kernel dependence. (L66-67)

• 77-79 - need reference(s) for these examples

Done (L79-80)

• 146 - Smith et al 2018 may not be the first ref for this - other studies used kernels for adjustments earlier (e.g. Vial et al 2013, Block and Mauritsen 2013)

Thanks for your suggestion. We have added the other two citations here (L145-146).

• 160 - define 'a control and a perturbed simulation' - it might not be immediately obvious to all readers

We've added clarification to this sentence (L161-164).

• 188 - 'holding the control and perturbed simulations' - should this be 'holding the input data of the control and perturbed simulations'?

We can see why this could be confusing and clarify what data is contained in the Xarray Datasets (L195).

• 241-242 - is the option to calculate all-sky or clear-sky feedbacks also applicable to other functions? If so, it would be worth mentioning it earlier in the first function which it is true for, or in the intro to section 3

We have revised the paragraph to note that the package computes the albedo and all the previously described feedbacks for clear-sky conditions (L146-147).

• 250-252 - while its great the kernels can be used for radiative adjustments too, I am not sure the cloud kernel will work for radiative adjustments with either the adjustment or residual method equations given in section 3.4 - this may be worth noting

These methods could theoretically be applied to fixed SST simulations to estimate the cloud rapid adjustments, but for the sake of simplicity, we leave this out of this section.

• 306 - strictly speaking, these are minima for the means, being negative not positive, but perhaps could be described as 'maxima in magnitude'

Thanks for catching this. We have revised the sentence to say "maxima in magnitude." (L317).

• 309-310 - would be good to suggest an explanation of how the inclusion of clouds or not might cause such a change in the spatial pattern of the mean temperature kernel

We now include an example to explain this sentence's meaning (L321-323).

• Figure 1 and figure 2 both skip the subfigure label 'g' and go straight to 'h'

We forgot the letter "g" in our alphabet when we created these figures. Thank you for pointing it out!

• 320 - avoid use of 'significantly' where not referring to statistical significance; 'larger' would suffice, or better yet quantify it

Thanks. We dropped the word "significantly" in this sentence.

• 329 - The numbering of the section would make more sense as 4.2 here rather than 4.1.1 (as there is presently no section 4.2, nor a 4.1.2 either)

 That was a mistake on our part. Thank you for catching it.

 • 333-334 - explain whether using the last 30 years of a 150 year simulation means that the surface T response has equilibrated sufficiently, or use a reference to state this is a sufficiently long enough length of time to use

Using the last ~30 years is common in polar amplification studies, so we added references to justify this choice. (L354-356)

• 363-364 - The description of figure 2j is inaccurate: the maxima over NE Pacific and the southern Atlantic and Indian Oceans are similarly large to the two mentioned in the text, and the contrast between land and sea is not as clear as between different land areas or different sea areas. I would suggest a different description.

Agreed. We have completely reworked this paragraph. (L384-391)

• Figure 2 - it would be helpful to separate the cloud feedback into LW and SW components, to match the treatment as in figure 3. This should help with the description of the spatial pattern in the text too. Perhaps this could be achieved without reducing the subfigure size (and still fitting on one page) by making it 4 columns instead of 2 columns of subfigures.

This is a great suggestion! Thank you!

• 372 - 'very large' - please quantify (especially because it is an interesting result!)

Done. Thanks! (L399-401).

• 404-405 - this is inconsistent with line 328, where you noted that the surface albedo kernel spread is indeed different between the all and clear sky in the tropics; I'm also not sure that this would prove the point that clouds are not the dominant cause of spread across kernels. I do agree however, in figure 1, that only the T kernel spread seems to differ a lot between all and clear sky - so maybe re-word this sentence if you want to make this point, or remove it.

After fixing the HadGEM2 surface albedo kernel you pointed out above, the clear-sky surface albedo standard deviation is less than half the size of the all-sky surface albedo kernel. We have changed L459-460.

• 408-410 - I am not clear how the global mean feedbacks in table 2 support the interpretation that the lapse rate or surface albedo feedbacks are most important for Arctic amplification?

This was indeed confusing and requires a deeper investigation to support these claims. We have reworked this sentence to align with our results (L463-465).

---

## Author Comment (AC2)

[revised manuscript text omitted]